# Dataset of Multi-Aspect Integrated Migration Indicators

Diletta Goglia * , Laura Pollacci and Alina Sîrbu

Department of Computer Science, University of Pisa, 56127 Pisa, Italy; laura.pollacci@unipi.it (L.P.);
alina.sirbu@unipi.it (A.S.)
* Correspondence: d.goglia@studenti.unipi.it

**Abstract:** Nowadays, new branches of research are proposing the use of non-traditional data sources for the study of migration trends in order to find an original methodology to answer open questions about cross-border human mobility. New knowledge extracted from these data must be validated using traditional data, which are however distributed across different sources and difficult to integrate. In this context we present the Multi-aspect Integrated Migration Indicators (MIMI) dataset, a new dataset of migration indicators (flows and stocks) and possible migration drivers (cultural, economic, demographic and geographic indicators). This was obtained through acquisition, transformation and integration of disparate traditional datasets together with social network data from Facebook (Social Connectedness Index). This article describes the process of gathering, embedding and merging traditional and novel variables, resulting in this new multidisciplinary dataset that we believe could significantly contribute to nowcast/forecast bilateral migration trends and migration drivers.

**Keywords:** international migration; migration flows; migration stocks; bilateral migration; Facebook social connectedness index; migration nowcasting; migration drivers

## 1. Introduction

Human migration is a complex phenomenon characterized by several related factors. It is as ancient as human history, and it has been widely studied, explored and described over time. The technological advancements and the rapid and drastic changes that society faced in the 21st century have impacted on the migration phenomenon, which consequently has undergone radical modifications. Taking into account this information about societal changes and technological progress (such as economic, cultural, and social big data) can be an effective strategy nowadays to detect new trends in bilateral migration and to better understand and nowcast it [1,2].

In the last years, the pursuit of original drivers and measures is becoming an increasing requirement to migration studies, considering the new methods and technologies used to characterize and understand the human migration phenomenon. Many researchers [3–14] have proposed to employ non-traditional data sources to study migration trends, including so-called Social Big Data such as online social networks. This unconventional approach is intended to find an alternative methodology to ultimately answer open questions about human migration (i.e., nowcasting flows and stocks, studying integration of multiple sources and knowledge, and investigating migration drivers). The new data have the advantage of timeliness and large geographical coverage, but also disadvantages in terms of selection bias and amount of resources required to process [3]. Therefore, models extracted from these data need to be carefully validated, typically with traditional data sources. In this context of meaningful data combination, many types of data exist, still

very scattered and heterogeneous including traditional data [15], making integration far from straightforward.

In this work we propose a dataset to be exploited in migration studies as a concrete example of this new integration-oriented approach: the Multi-aspect Integrated Migration Indicators (MIMI) dataset. It includes both official data about bidirectional human migration (traditional flow and stock data) with multidisciplinary variables and original indicators, including economic, demographic, cultural and geographic indicators, together with the Facebook Social Connectedness Index (SCI).

Thanks to this variety of knowledge, experts from several research fields (demographers, sociologists, economists) could exploit MIMI to investigate the trends in the various indicators, and the relationship among them. Moreover, it could be possible to develop complex models based on these data, able to assess human migration by evaluating related interdisciplinary drivers, as well as models able to nowcast and forecast traditional migration indicators in accordance with original variables, such as the strength of social connectivity. Here, the SCI could have an important role. It measures the relative probability that two individuals across two countries are friends with each other on Facebook, therefore it could be employed as a proxy of social connections across borders, to be studied as a possible driver of migration. The combination of this index with socioeconomic variables measuring the similarity of two locations (such as per capita income, religiosity and language) already appeared in [16,17] where it has been shown that pairs of locations that are more similar on these dimensions share more friendship links. Even though the Facebook SCI suffers from the same selection bias as many other online social networks, it has the advantage that it is calculated directly by Meta, therefore, includes all data and not a subsample. For the same reason, it is easy to use since values are already calculated from the large micro-level data.

All in all, the motivations for building and releasing the MIMI dataset lie in the need of new perspectives, methods and analyses that can no longer prescind from taking into account a variety of new factors. The heterogeneous and multidimensional sets of data present in MIMI offer an all-encompassing overview of the characteristics of human migration, enabling a better understanding and an original potential exploration of the relationship between migration and non-traditional sources of data.

## 2. Background and Motivation

### 2.1. Definitions of Terms

While there is no formal legal definition of an international migrant [18], the United Nations Department of Economic and Social Affairs (UN DESA) [19] defines "international migrant" as someone who moves away from his or her country of usual residence (defined by UN DESA as "the place where a person lives and where he or she normally spends the daily period of rest" [19]). Generally, a distinction is made between short-term or temporary migration, covering movements with a duration between three and 12 months, and long-term or permanent migration, referring to a change of country of residence for a duration of one year or more. In this work, we will use the term "migrant" indicating a long-term international migrant as defined by UN DESA.

We rely on the definitions provided by the Migration Data Portal [1] distinguishing migration data in *stocks* and *flows*. The former term represents the absolute number of migrants in a country at a given time [18], while the latter is a dynamic measure counting the number of people who have moved countries in a given time period, i.e., the number of migrants who have crossed an international border. Since migration flows can both consider people arriving in a specific country or departing from it [20], we will also refer to *immigration* or *emigration* flows, and so to *immigrants* and *emigrants*. While migration stocks only refer to people who have changed their usual residence, flows can be measured counting migrants *by citizenship* or *by residence*.

Despite the various definitions provided in literature to the term of Social Big Data (SBD) [21–23], we believe that the most suitable for the framework of this work is the

following: "Big Social Data is any high-volume, high-velocity, high-variety and/or highly semantic data that is generated from technology-mediated social interactions and actions in the digital realm, and which can be collected and analyzed to model social interactions and behavior" [24]. This conceptualization of SBD perfectly suits the framework of this work, as well as the purposes for which we intend to use SBD: it synthesises the various features of this type of data, pointing out its heterogeneous nature and unavoidably linking it to this new sociability implemented through technological tools. In this work we refer to Online Social Networks (OSNs) [21], and in particular to Facebook.

Finally, with "data integration" we refer to the process of combining data residing at different sources in order to provide a unified view of these data [25]. This unified view, besides facilitating access and exploration, can offer an excellent starting point for subsequent analyses. "Combining" represents a non-trivial process, since collecting data from all the different sources is not enough: it is necessary to model and unify data semantics and architecture in order to produce a consistent and homogeneous final fruition. The problem of data integration appears more and more frequently, in the same way that the volume of this data and the need to unify existing sources in a coherent and meaningful way is exploding. Heterogeneity and sparseness of data make this process even more complex.

This work, as well as the data included in the MIMI dataset, provide knowledge regarding the concepts and domains presented above.

### 2.2. International Statistics in Migration Studies

Today more than ever, migration is at the centre of the issues of the 21st century [26]. The 2030 Agenda for Sustainable Development Goals [27]—adopted in 2015—is the first international framework to recognise migration as a development dimension [15]. Thus, the new global development framework has posed enormous challenges to national statistical offices on how to interpret concepts linked to migration, retrieve, integrate and explore the sources of information, and develop migration indicators and policies. To understand and capturing migration trends and global migration patterns, develop scenarios, and design and support effective evidence-based migration policy, more timely, granular, timely, accurate and reliable data is required [15,26,28]. Considerable efforts have been made to structure international data systems and conceptualise, identify and bridge the gaps within and between them [3,15,29–33]. Efforts have led, to some extent, to improvements in the availability, quality, and comparability of data on international migration [15]. Several countries and international organisations, e.g., the United Nations (UN), United Nations Department of Social and Economic Affairs (UN DESA), Eurostat (ESTAT), Organisation for Economic Cooperation and Development (OECD), International Organisation for Migration (IOM), the World Bank, and Non-Governmental Organisations (NGOs) as UNICEF, collect data on international migration. However, despite the attention of national states to the importance of statistical data on migration, gaps and incompatibility caused—primarily—by the use of national definitions, often not compatible with the UN's 1998 recommended standards [34], remain crucial challenges for integrating these data.

In a recent study, gaps in international statistics are analysed by Ahmad-Yar and Bircan [15] with respect to five categories, including *(a)* definitions and measures, *(b)* drivers or reasons behind migration, *(c)* geographic coverage, *(d)* gaps in demographic characteristics, and *(e)* the time lag in the availability of data. Based on this classification, the review, conducted on a set of about 80 research products, underlines that in the more than 900 cases of the gaps discussed in the literature found, 611 referred to the definitions and measures, 113 to demographic characteristics, 101 to drivers and reasons, 48 to geographic coverage and 43 to the timeliness of the data.

Despite gaps and limitations, many researchers and policymakers rely on official statistics, such as census and surveys, to study the migration phenomenon [3]. Census and surveys provide socio-demographic information and indicators about the population, including migrants. Census data are collected once in five or ten years, depending on the country, and provide—as migration-related information—citizenship, country of birth, last

place of residence, and the length of stay. In contrast with censuses, surveys are conducted based on primary purposes such as households, the labour market, and the community. These can also collect information as flows and stocks of migrants. Still, often there are few questions related to migration, and data retrieved refers to a small subset of the entire population. In addition to census and surveys, administrative data could be retrieved from registries and collected information about asylum seekers and refugees. Registry data are often detailed and are less costly than official statistics. However, the geographical resolution poses challenges in harmonising and comparing data from different countries. Administrative registers are often unsuitable for cross-country comparisons, and despite being detailed, they do not include irregular migrants and displaced people. As a result of this context, data concerning well-defined demography-related events such as births and deaths typically have good quality, at least in more developed countries where they are routinely collected [15,33]. These events are pretty exact and carefully recorded by administrative bodies and national statistical offices also for intra-national security reasons. However, the spatial coverage is limited. Conversely, the quality of data on migration, including population could stocks and flows, suffers from poor registrations [15,33]. Despite this, to date, the availability of stock data in developed countries is generally good, although not always detailed by demographic information, e.g., age distribution.

Besides collecting data and producing statistics, the dissemination of the latter also poses crucial aspects. Dissemination can occur through multiple channels and resources and with substantial differences in availability and quality. An aspect connected with quality is the documentation of the data, which, when present, is often inadequate, e.g., when metadata on data coverage and definitions are not provided [15,31,35].

All in all, we can conclude that internal and international migration data are usually challenging to find, obtain and harmonise [33]. Our work aims to fill the gaps in these directions, finding and integrating data from multiple traditional sources.

Sources for International Migration Statistics

The statistics available for research and end users are typically developed by processing raw collected data by the national statistical office and entities responsible for statistics production. As previously discussed, several international organisations and governments gather data on migrants.

Among the international organisations we mention:

**Eurostat**. The Statistical Office of the European Union, Eurostat, collects and processes data from the Member States of the European Union. Eurostat provides annual statistics, flows, demographics and projections. The information collected includes requests for asylum and residence permits, also about minors. Along with statistics on international migration, Eurostat also provides data on the integration of migrants based on rates of education, health, employment, and social inclusion.

**UN DESA**. The UN DESA data is derived from population censuses, population registers, and national surveys of the UN member states. Estimations on international migrant stocks are provided every five years from 1990 to 2015 plus 2019 for all countries and areas of the world. For the countries that not collect migration data, the UN estimates the statistics based on all international migration stocks, age, sex, origin, and destination. Data follows a classification based on countries' development [2] and include foreign-born and foreigns, demographic characteristics, e.g., age group and gender, and origin and destination-based data.

**IOM**. IOM generically collects data during support missions on migrant border crossings and emergencies. In 2015, the IOM established the Global Migration Data Analysis Centre (GMDAC), which hosts the Migration Data Portal [36], a platform providing access to comprehensive and reliable migration statistics useful for researchers and policymakers.

**OECD**. The OECD provides three different databases on migration data: the OECD international migration database, the database on immigrants in OECD countries, and indicators of immigrant integration. The OECD international database provides annual flows,

stocks, and nationality acquisition by foreign-born and foreigners in OECD member states. Data also include information on asylum seekers and foreign-born stocks. The database of immigrants in the OECD provides comparative information about the demographic and labour characteristics of immigrants living in OECD countries. Further, the indicators of the immigrant integration portal give a set of integration indicators in fields such as employment, education, skills, social inclusion, and cohesion.

**UNHCR**. The United Nations High Commissioner for Refugees (UNHCR) collects data regarding refugees and vulnerable groups, including also asylum seekers, displaced persons, and returned migrants.

### 2.3. Novel Sources and Types of Data

Beyond the gaps and incompatibility, when carried out, data collection generally has very high costs for governments and international organisations and often does not capture spatial and temporal dimensions of the migration phenomenon [37]. As a consequence of all the challenges, in the last decade, the potential of novel sources and types of data are being assessed to fulfil gaps and overcome limitations in understanding the migration phenomenon and for policy-making [3,28,37–40]. The literature has pointed out that these novel data (sources) have a lot of potential, e.g., for developing countries where official data is incomplete or is not available, but at the same time, it poses challenges that concern both technical and ethical issues [3,28,38–40]. Although non-conventional data differ significantly from traditional ones in the purpose of data collection, data heterogeneity, time and spatial coverage, and completeness, as shown in [28].

Big data sources such as—but not limited to—social media, satellite data, internet services, mobile phones, air traffic, purchase transactions, and money transfers attracted growing attention. These data contain detailed information about individuals and typically cover large sets of populations.The literature has pointed out that the reliability of data could be compromised due to bias of users' characteristics in the sample [3]. For instance, 65% of Facebook's users are under the age of 35, with high rates of users with 30–35 years and 36/45 years (98% and 78%, respectively [3]. Nevertheless, in comparison to official/traditional data, the main advantages of so-called *non-conventional* (or *innovative*) data sources lie in their greater geographic and temporal granularity, (almost) real-time availability, and coverage—even potentially worldwide—which allows more immediate comprehensive comparisons than traditional data sources, which are often limited in sample size, extension, and updates [3,28].

In some cases, innovative data most extensively used to fill in the gaps in traditional statistics have shown the potential to overcome official and survey data. To date, an even growing literature is adopting mixed methodologies based on integrating traditional and non-conventional data sources [3,28,39]. In recent years Facebook data has proved to be a precious source of information for multiple purposes. Data, aggregated and anonymised, has high geographical and temporal resolution and cover a large set of the population. However, the differences in the distribution of users by age, gender, and origin need to be taken into account [41]. Data for Good at Meta (see Section 2.4) is a data-sharing initiative that provides anonymised and aggregated datasets derived from multiple sources, including Facebook, to address humanitarian challenges and empower social good.

### 2.4. Migration Indicators

Despite challenges and limits posed by data collection and integration, several indicators have been developed using these to understand, provide and compare measurable information on the migration phenomenon [3,28,38,39]. According to the differences in statistics and data, a wide range of indicators and indexes has been developed, which vary in temporal, geographical, and thematic coverage. The proliferation of indicators has heterogeneously focused on a few areas of migration research, such as mobility and policy-making. However, also in this latter context, admission citizenship, acquisition-related policies, and, more recently, integration policies seem to attract more interest than the

return, emigration, and diaspora policies, which are still overlooked. The 2021 IOM Global Migration Indicators [42] report summarises recent migration trends based on periodically updated data from the Global Migration Data Portal. Here, migration indicators are classified into twenty groups, including, among others, migrant population, stocks, and flows, labour migration, remittances, forced migration, displacement and resettlement, and Big Data for migration [4]. Population Division of UN DESA [43], in collaboration with UNICEF, provided the Migration Profiles Common Set of Indicators. Profiles, documented with country-related definitions, provide *(a)* population indicators, e.g., population estimates and projections, effects of future international migration on the total and working-age population, *(b)* development indicators, e.g., life expectancy, GDP per capita, and remittances; and *(c)* international migration indicators by demographic and geographical constraints, e.g., major age-group and country of origin and destination, and refugee populations by country of origin and destination, among others. Further, according to international migration data, OECD provides key indicators [44], including countries' in and outflows, asylum seekers, and foreign-born stocks.

As a result of the most recent research, the initiatives to harness the potential of unconventional and innovative resources and data have grown significantly. These several initiatives, referred to as *Data for Good* initiatives [28,45], differ considerably from each other and focus on numerous aspects including indicators. Among the 11 categories identified in [45], dataset providers, data governors, and data strategy providers' initiatives—together with community builders one—seem to favour migration studies and human mobility (and demography) the most [28]. *Dataset provider initiatives* [5] include "Data for Good at Meta" [6], based on data from Meta platforms and other resources, provides *(a)* anonymised and aggregated data and maps regarding population, mobility (including displacement), connectivity and infrastructure, economic and poverty, forecasts, and social connections, e.g., Social Connectedness Index; *(b)* surveys; and *(c)* statistics. The Social Connectedness Index (SCI) [7] measures the strength of connectedness between two geographic areas as represented by Facebook friendship ties, according to Bailey et al. [17,46]. Further, Tjaden et al. [47] studied the potential of SCI with UNDESA to predict international flows.

Under these conditions, one of the main purposes of our introduction of the MIMI dataset lies in the integration of bidirectional human migration indicators, including traditional flow and stock data, multidisciplinary variables and original indicators, including economic, demographic, cultural, and geographic indicators from traditional well-established sources, namely UN and ESTAT, and last but not least, Facebook SCI data. Among the numerous innovative indicators, the literature has shown that Facebook Social Connectedness Index can be used to investigate the relationships between digital social connections, migration patterns, and social, cultural, and demographic characteristics [16,38,46]. On the one hand, we aim to contribute to integrating traditional and innovative data to facilitate their joint use by maximizing the potential of the resources available today. On the other hand, we believe that the SCI, in particular, can add an additional level of information to traditional indicators. The SCI measures the relative probability that two individuals across two countries are friends with each other on Facebook. Thus, it could be employed as a proxy of social connections across borders to be studied as a possible driver of migration. Thanks to this variety of knowledge, experts from several research fields, e.g., demographers, sociologists, and economists, could exploit MIMI to investigate the trends in the various indicators and the relationship among them.

All in all, the motivations for building and releasing the MIMI dataset lie in the need for new perspectives, methods, and analyses that can no longer be prevented from considering various new factors. The heterogeneous and multidimensional sets of data present in MIMI offer an all-encompassing overview of the characteristics of human migration, enabling a better understanding and an original potential exploration of the relationship between migration and non-traditional sources of data.

## 3. Data Description

The MIMI [48] dataset (May 2022) was released under the Creative Commons Attribution 4.0 International Public License (CC BY 4.0 [8]) and is publicly available on Zenodo (https://doi.org/10.5281/zenodo.6493325). It contains more than 28,000 records and 870 different variables. In this section we provide all the dataset specifications and describe how each variable was built.

### 3.1. Data Structure

#### 3.1.1. Data Files and Format

The MIMI dataset is made up of one single CSV file that includes 28,821 rows (records/entries) and 876 columns (variables/features/indicators). Each row is identified uniquely by a pairs of countries, built from the joining of the two ISO-3166 alpha-2 codes for the origin and destination country, respectively. The dataset contains as main features the country-to-country bilateral migration flows and stocks, together with multidisciplinary variables measuring cultural, demographic, geographic and economic variables for the two countries, together with the Facebook strength of connectedness of each pair.

#### 3.1.2. Geographical Coverage

The dataset covers 255 different countries belonging to the following macro-areas: North America, South America, Europe, Asia, Africa, Oceania, Antarctica. Not all country pairs are present, due to missing data in the original sources.

#### 3.1.3. Temporal Coverage

Our work focuses on the integration of various indicators related to migration, including Facebook Social Connectedness, therefore the choice of the time range has been calculated accordingly. Our intention was to build a dataset that could also be useful for the study of the differences between contemporary and past trends (e.g., alterations of some phenomenons, consistent changes of values compared to the past, consequences of previous data on the last few years, etc.). For this reason the time coverage starts from the year 2000, and goes up to 2022. Some projections also cover future years, up to 2025. Certainly, data selection according to predetermined temporal ranges always depends on the availability of sources for each variable considered. For example, during our data collection phase, Eurostat was not providing information about population density of countries before 2008. Facebook data is only available for August 2020 and October 2021. Table 1 provides a detailed description of the temporal coverage of each type of variable.

**Table 1.** Temporal coverage of each variable. "End" always refers to the latest available measure. For all the abbreviations refer to Section 6.

| *Yearly Measures* | | |
|---|---|---|
| **Start** | **End** | **Variable** |
| 2000 | 2020 | GDP at PPP |
| 2000 | 2022 | UN population |
| 2008 | 2019 | EUROSTAT population density, EUROSTAT total immigrants and emigrants for each country |
| 2010 | 2019 | EUROSTAT migration flows |
| 2010 | 2020 | UN migration flows, UN total immigrants and emigrants for each country |
| 2010 | 2021 | EUROSTAT population |
| *Five-Year Measures* | | |
| **Start** | **End** | **Variable** |
| 2000 | 2020 | Migration stocks |
| 2000 | 2025 | NET migration and NET migration rate |
| *Snapshots* | | |
| **Date** | | **Variable** |
| August 2020 October 2021 | | Facebook SCI |

### 3.2. Dataset Variables

#### 3.2.1. Variable Definition

In this section we are going to list all the indicators included in the MIMI dataset, then we will describe them in detail in the following section and in Appendix A. Table 2 contains a complete definition of all variables, grouped and categorized by context ("variable area"). The column "Name" contains the identifier of each variable. Since it would not be possible to list all variables, we develop a naming rule in order to include them all in the table. Specifically, each name has an *invariant* part, always formatted with *italic* font, and a `placeholder` part, always formatted with `monospace` font, that can take different predefined values, as follows:

- `country` should be replaced with *origin* or *destination*.
- `year` and `start-end` should be replaced, respectively, with the reference year (in case of annual variable) or reference year range (for NET migration and NET migration rate). Substituted values should be consistent with the temporal coverage available for each indicator, which can be found in Table 1.
- `source` allows *UN* and *ESTAT* as replacement values, corresponding to United Nations and Eurostat data sources.
- `sex` should be substituted with *F*, *M* or *T* (respectively, female, male or both).
- `age` allows only *T* as replacement value for data obtained from UN (both flows and stocks), while it can take four different values for ESTAT flows: *T* (total), *<15* (less than 15 years), *15–64* (from 15 to 64 years), *>65* (65 years or over).
- `by` should be substituted with *cit* or *res* for selecting, respectively, migration flows or total migration by citizenship/birthplace or by residence. More details can be found in the following section, related to description of *variable 32*.

From this simple rule it is possible to derive the exact name of each single indicator. Some examples are provided in the footnotes of Table 2.

**Table 2.** Variable list. The exact name of the each single indicator can be retrieved by following the rule explained in Section 3.2.1.

| Variable Area | | Name | Related to | Brief Description | Dtype |
|---|---|---|---|---|---|
| Index | 1 | *from_to* | | Unique identifier of each record | object |
| Facebook | 2 | *sci_*`year` | Pair of countries | Social Connectedness Index | float64 |
| | 3 | *from_to_cont* | | Pair of continents codes | object |
| | 4 | *geodesic_distance_km* | | Distance between countries | float64 |
| | 5 | `country`*_country* | | ISO-3166-1 alpha-2 code | object |
| | 6 | `country`*_name* | | ISO-3166 name | object |
| | 7 | `country`*_alpha_3* | | ISO-3166-1 alpha-3 code | object |
| | 8 | `country`*_official_name* | | ISO-3166 official name | object |
| | 9 | `country`*_cont_code* | | Continent code | object |
| Geographic | 10 | `country`*_cont_name* | | Continent name | object |
| | 11 | `country`*_latitude* | | Centroid latitude | float64 |
| | 12 | `country`*_longitude* | | Centroid longitude | float64 |
| | 13 | `country`*_coordinate* | | Centroid *(lat, long)* pair | object |
| | 14 | `country`*_neighbors* | | List of bordering countries | object |
| | 15 | `country`*_area* | | Area (in squared kilometers) | float64 |
| | 16 | `country`*_religion* | | List of religions | object |
| | 17 | `country`*_gdp_*`year` | Single country | Annual GDP at PPP | float64 |
| | 18 | `country`*_languages* | | List of spoken languages | object |
| | 19 | `country`*_fb_users* | | Number of Facebook users | Int64 |
| | 20 | `country`*_fb_users_perc* | | Percentage of Facebook users | float64 |
| Interdisciplinary | 21 | `country`*_PDI* | | | Int64 |
| | 22 | `country`*_IDV* | | | Int64 |
| | 23 | `country`*_MAS* | Cultural Indicators | | Int64 |
| | 24 | `country`*_UAI* | | | Int64 |
| | 25 | `country`*_LTO* | | | Int64 |
| | 26 | `source`*_*`country`*_pop_*`year` [1] | | Annual population stocks | Int64 |
| | 27 | `source`*_*`country`*_pop_dens_*`year` | | Annual population density | float64 |
| | 28 | `source`*_*`country`*_total_imm_*`by`*_*`year` | | Annual total immigrants | Int64 |
| | 29 | `source`*_*`country`*_total_em_*`by`*_*`year` | | Annual total emigrants | Int64 |
| Demographic | 30 | `source`*_*`country`*_net_migr_*`start-end` [2] | | Five-year NET migration | float64 |
| | 31 | `source`*_*`country`*_net_migr_rate_*`start-end` | | Five-year NET migration rate | float64 |
| | 32 | `source`*_*`year`*_*`sex`*_*`age`*_*`by` [3] | Pair of countries | Annual migration flows | Int64 |
| | 33 | `source`*_migr_stocks_*`year`*_*`sex`*_*`age` [4] | | Five-year migration stocks | Int64 |

[1] e.g., *ESTAT_origin_pop_2017*. [2] e.g., *UN_destination_NET_migr_2005-2010*. [3] e.g., *ESTAT_2015_M_>65_res*. [4] e.g., *UN_migr_stocks_2000_F_T*.

3.2.2. Variable Description and Sources

In this section we briefly describe the indicators included in the dataset, while in Appendix A we specify in detail each single variable listed in the previous section, also reporting all the data sources: some indicators may have multiple sources since they were necessary to better integrate missing values.

Besides the **Index** (*variable 1* in Table 2), which uniquely identifies each record representing pairs of countries, the MIMI dataset includes:

- **Facebook data** (*variable 2* in Table 2). This indicator represents the non-traditional variable within the context of migration studies that we included. It consists of the so-called Facebook **Social Connectedness Index** [9] publicly provided by the "Data for Good at Meta" [10] organisation on the "Humanitarian Data Exchange, Data for Good" platform [11].We include in the MIMI dataset two snapshots available at the time of writing, which are related to August 2020 and October 2021 [49]. Country-to-country values of SCI (both for 2020 and for 2021) are available for more than 34,000 pairs, so that to cover 61% of all the pairs of countries included in the whole MIMI dataset.
- **Geographical variables** (*variables 3–15* in Table 2). These variables portray and contextualise both origin and destination countries at geographical level, providing all the necessary information to describe them, starting from the official codes and names, up to their land extent and how far they are from eachother.
- **Interdisciplinary indicators** (*variables 16–25* in Table 2). Some of these indicators are considered non-traditional data in the context of migration studies since their use in migration understanding and nowcasting is poorly documented in literature. Despite this, most of the available studies consider these variables as relevant, as they are related to international migration trends. They include economic variables, cultural variables (related to religion, language and cultural traits) and measures of Facebook users.
- **Demographic variables** (*variables 26–33* in Table 2). These variables correspond to traditional migration and population measures obtained from official statistics, either from national censuses or from the population registries.

## 4. Methods

The process of building the MIMI dataset was implemented in Python 3.8, with the aid of the Jupyter 6.0.3 software, part of the IPython open source project of the python community [12]. We started with **data collection and acquisition**, by exploring open source portals, and selecting and downloading the data.

We built three separate initial datasets, that we could reuse in different contexts where needed. The first and the second included, respectively, migration flows imported from EUROSTAT and migration flows and stocks imported from UN. The third consisted of geographic, demographic and interdisciplinary variables related to single countries, (*variables 5–25* in Table 2).

Then a **pre-processing** phase started, where we carried out data understanding, cleaning and preparation. This has been managed by defining some functions that automatically clean and prepare the source datasets. Here our data was subjected to various computational standard processes (such as outlier detection, duplicate handling, uniforming notation, etc.). Some of the operations that have been performed at this level included the selection of task-relevant data (detection of country-to-country valid records, aggregation removal, and non-bilateral flow elimination). *Variables 32, 33* were also subjected to a **data transformation** phase, to reshape the data in order to resemble the final structure. Concretely, this meant converting, grouping, and unstacking records of source datasets in order to transform them in variables (columns) corresponding to country pairs.

The following step was **data integration** were the three dataset were joined together into a large matrix, matching single countries or pairs when needed. Once the latest variables (*2–4* in Table 2) have been added and the integration was completed, it was

helpful to check data semantics and statistics of the resulting dataset, in order to verify the need for a further cleaning step.

The final **data quality assessment** phase was one of the longest and most delicate, since many values were missing and this could have had a negative impact on the quality of the desired resulting knowledge. The missing values have been integrated from additional sources, as reported, for each variable, in Section 3.2.2.

The entire dataset building process is summarised in Figure 1.

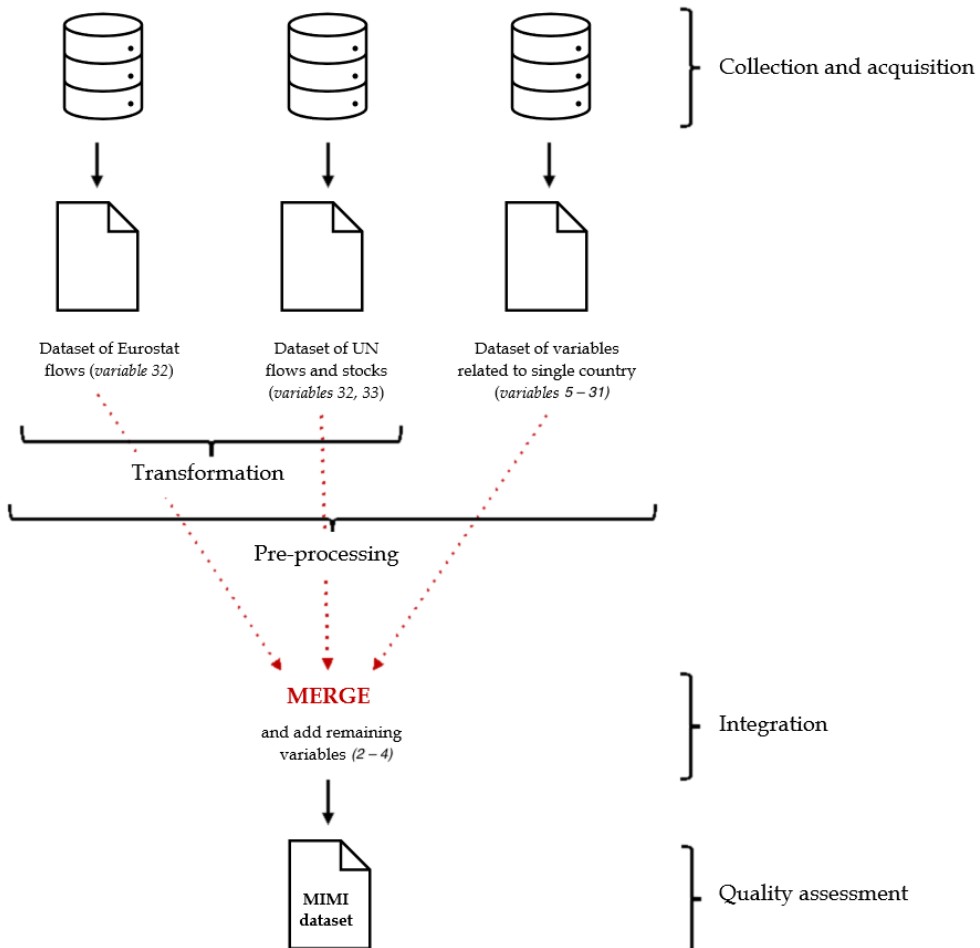

**Figure 1.** Pipeline of the process of building the MIMI dataset.

## 5. Use Case Examples

In this section our focus is on documenting and describing salient patterns in distributions and correlations of the various indicators included in the MIMI dataset. We do not seek to provide causal analyses, nor do we want to imply causal relationships at this stage: however we believe it can be useful to analyse the obtained numerical results since they may guide possible future research and lead to interesting progress in human mobility studies.

Unless otherwise specified, correlation values have been computed as Pearson's correlation [50], measuring the linear relationship between two variables: values of −1 or +1 imply an exact linear relationship, while 0 implies no correlation. We also used Spearman rank-order formulation [51] to find non linear monotonicity between variables, as well as Kendall's tau-b formulation [52] to check ranking correspondence. Similar to Pearson's correlation, Spearman's varies in the same range [−1, +1] with 0 implying no correlation. For Kendall's correlation, values close to the positive or to the negative bound indicate, respectively, strong agreement or disagreement of rankings.

*p*-values have been computed in order to confirm of refute the relevance of each correlation value: results are indicated in heatmaps with a number of asterisks proportional to the relevance obtained:

| | | |
|---|---|---|
| no asterisks | no relevance | *p*-value $\geq 0.05$ |
| * | little relevance | $0.01 \leq$ *p*-value $< 0.05$ |
| ** | medium relevance | $0.001 \leq$ *p*-value $< 0.01$ |
| *** | high relevance | *p*-value $< 0.001$ |

### 5.1. Data Statistics, Distributions and Correlations

In this section we provide some practical examples of how to explore the data. A Github repository (https://bit.ly/GitHubMIMI (accessed on 21 August 2023)) contains source code of examples of how to read and analyse the data. All the examples are provided in Python 3 language.

We start with some characteristics of the SCI, in Figures 2–4. Figure 2 shows the distribution of the indicator in two snapshots available. We observe that most values are lower, while some countries have some very large values. We also note that 2021 values seem to be lower than those in 2020. In Figure 3 we shot 99th percentile of SCI values, i.e., countries with largest connection strength. We note that the top pairs correspond to small island countries (e.g., the top are Micronesia and Guam). This means that the probability of selecting two persons connected on Facebook from these regions is high, and this could also depend on the fact that those countries are rather small, and isolated in the ocean. Figure 4 shows aggregated SCI (averages) between continents. Very high values of connectivity between different continents can be noticed for North and South America. Intra-continental connections are always much stronger than inter-continental connections: this could support what stated in [16] about the intensity of friendship links strongly declining with the geographic distance.

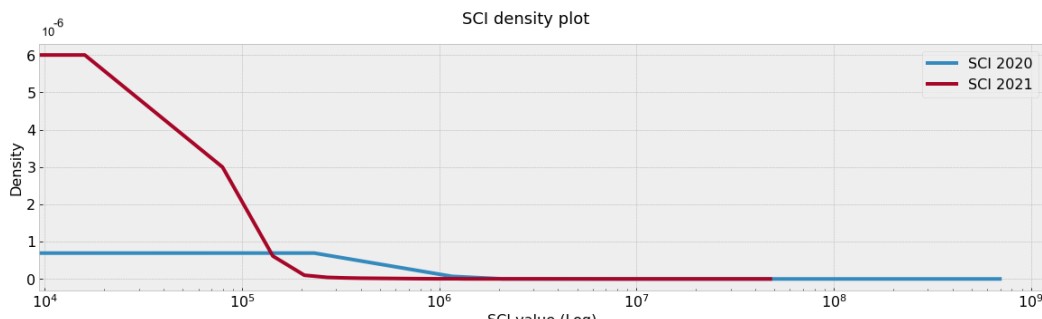

**Figure 2.** Density plot of SCI 2021 with logarithmic x axis. It shows a strongly right-skewed distribution, meaning that the smallest values of the indicator are the most frequent.

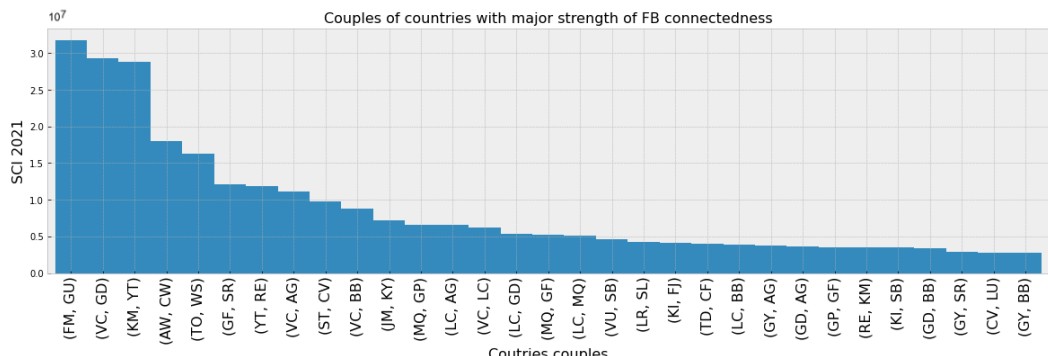

**Figure 3.** Sample of the highest value of SCI 2021, over the 99th percentile. It displays country pairs with the highest strength of Facebook connectivity.

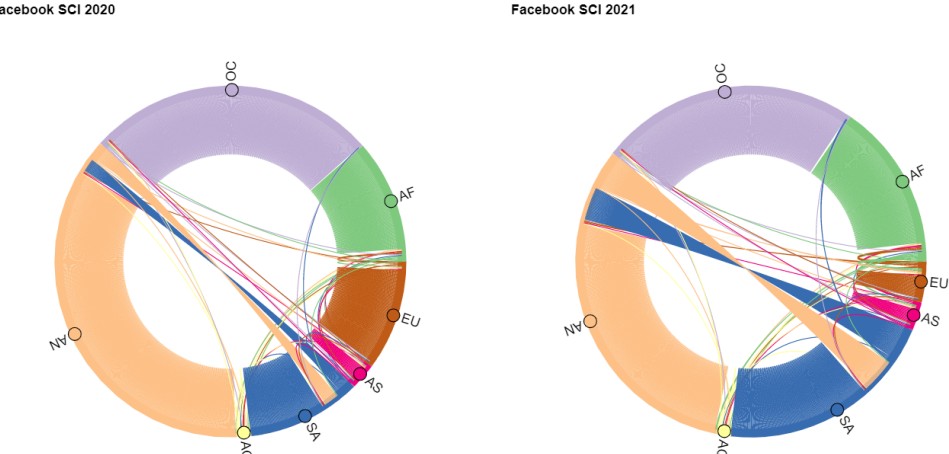

**Figure 4.** Facebook strength of connectedness among continents (aggregation of averaged SCI for each couple of countries). The colours correspond to the residence continent of the persons, while the chords show to which continents the residents of one continent are connected with.

We illustrate bidirectional international migration in 2018 (both by citizenship and by residence) between pairs of continents in Figure 5, where values are aggregated as the total sum of migration flows shared by pairs of countries in the given continent. We also include the years 2010 and 2014 in Appendix B. From Eurostat data, which has low resolution outside Europe, we observe that internal European flows are quite large, with a strong immigration from Asia to Europe, followed by South America and Africa. UN flows on the other hand include all continents, with Europe less represented. We note strong flows between Asia and North America, as well as between North America and South America. We also see large internal migration in Africa.

We zoom in to observe individual country pairs in Figure 6 (2019 EUROSTAT flows), Figure 7 (2020 UN flows) and Figure 8 (2020 UN stocks). Almost all the pairs of countries reported in Figure 8 are included in the "top 20 international migration country-to-country corridors, 2020" list in the World Migration Report 2022 [53], (e.g., Mexico—United States, Syria—Turkey, India—Saudi Arabia, United Arab Emirates and United States, Afghanistan—Iran, Myanmar—Thailand), meaning that the greatest communities of permanently residing migrants in a host country (international diaspora) have developed over years for safety reasons. A consistent flow of returning citizens can be noticed for Thailand in Figure 7, probably due to COVID-19, since in 2020 the pandemic prompted the return of hundreds of thousands of migrants to their countries of origin [54]. Regarding migration stocks in Figure 8, the impact of COVID-19 on the global population of international migrants is difficult to assess, since the latest available data refers mid-2020, fairly early in the pandemic. However, it is estimated that the pandemic may have reduced the growth in the stock of international migrants by around two million [53,55].

Figure 9 explores the changes in trend of NET migration rate for a small sample of countries. We note countries such as France, Italy, Spain, the United Arab Emirates, the United Kingdom, that have been important migration attractors over the years, with positive net migration rates. Conversely, some countries such as Cuba, Egypt, India, Poland, Romania, have consistently been the origin country for migration over the years. Changes in trends are visible over time, with many of the example countries tending to get closer to the 0 line, i.e., an inversion of trend.

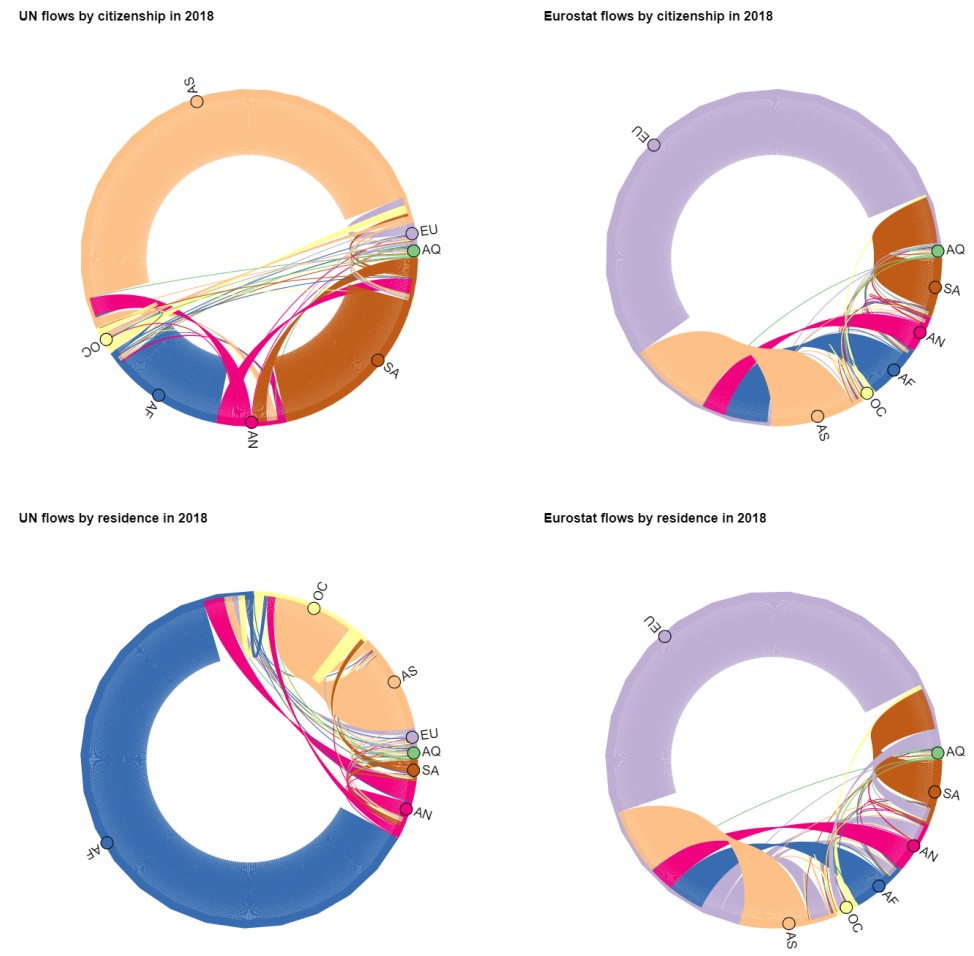

**Figure 5.** Inter-continental migration flows by citizenship and by residence from UN (**left**) and EUROSTAT (**right**) in 2018. The colours correspond to the citizenship (**top**) or residence (**bottom**) continent of the migrants, while the chords show to which continents the citizens/residents of one continent migrate to.

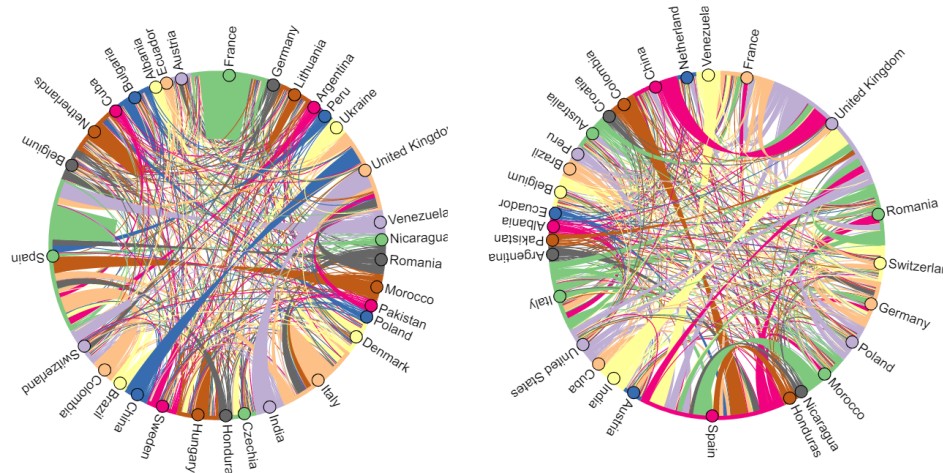

**Figure 6.** EUROSTAT bilateral migration flows by citizenship (**left**) and by residence (**right**) in the most recent year available (2019): pairs of countries with the highest numbers of migrants. The colours correspond to the citizenship (**left**) or residence (**right**) country of the migrants, while the

chords show to which countries the citizens/residents of each country migrate to. It is evident that, in both cases, countries receiving the highest number of immigrants are United Kingdom, Italy and Spain. A consistent flow of returned migration can be noticed for both France and Spain.

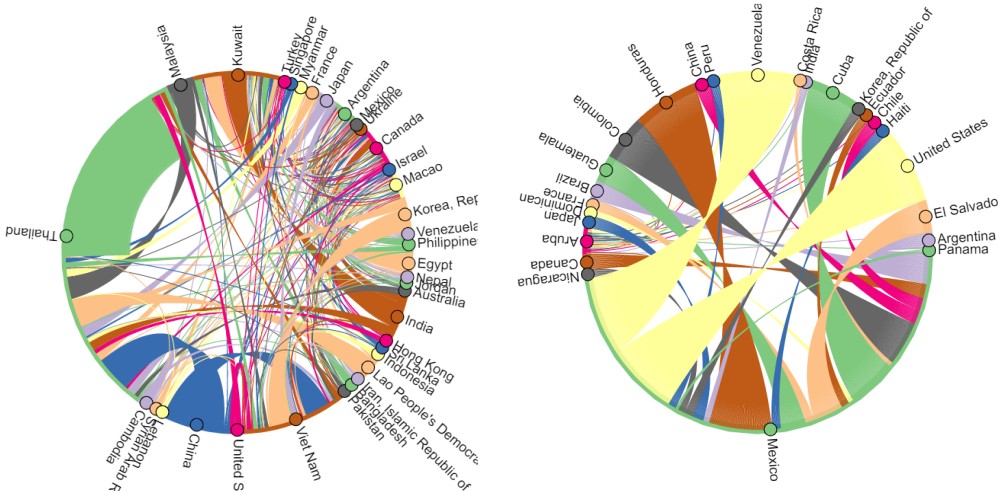

**Figure 7.** UN bilateral migration flows by citizenship (**left**) and by residence (**right**) in the most recent year available (2020): pairs of countries with the highest numbers of migrants. The colours correspond to the citizenship (**left**) or residence (**right**) country of the migrants, while the chords show to which countries the citizens/residents of each country migrate to. Besides welcoming a large number of international migrants, Thailand is also facing a huge returning migration. High immigration (by residence) is also concerning Mexico.

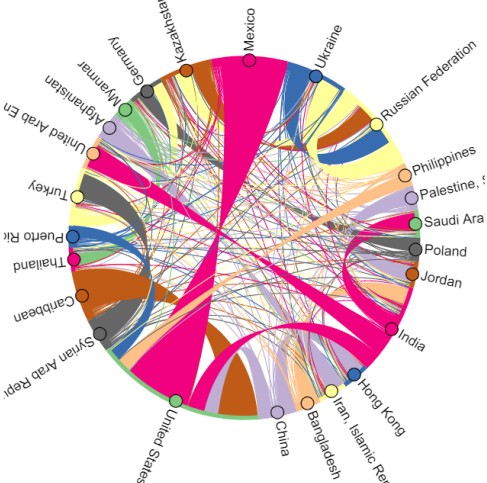

**Figure 8.** UN bilateral migration stocks (international diaspora) in the most recent year available (2020): pairs of countries with the highest numbers of permanent residing migrants. The colours correspond to the origin country of the migrants, while the chords show in which country the stocks of a certain origin are reported. US is hosting many voluminous stocks of migrants coming from different nations (mostly from Mexico). On the contrary, substantial stocks of Indian migrants can be found in Saudi Arabia, United Arab Emirates and USA.

The boxplots in Figure 10 display the statistical distribution of migration flows and stocks values over the years, divided by sex, for EUROSTAT flow data. Statistic evidence on male migration always reveals larger numbers with respect to female migration (about gender dimensions on human migration refer to [56]). Increasing migration trends over time are clear from the timeseries data of EUROSTAT flows. Similar observations can be made for UN flows and stocks (shown in Appendix B, Figures A3 and A4).

The heatmaps in Figure 11 show correlations between the computed ratio of total migrants and total population of a country and its cultural indicators. Since these latter have been collected in 2008 we correlated them with the closest possible year available in our migration data, which corresponds to 2010. For UN data by citizenship, both immigration and emigration percentages show negative correlations with IDV and MAS indicators (*variables 22, 23* in Table 2) which may suggest that the more a country faces with migration phenomenon, the less it is inclined to individualistic behaviours and to those "masculine societal values" described in Appendix A. Emigration appears to be negative correlated with LTO but positively correlated with UAI (respectively, *variables 25* and *24* in Table 2) indicating weaker projection in the future and weaker tolerance for news and changes for those countries where the emigration is higher. For EUROSTAT data by residence, instead, we can observe an opposite trend of correlation for UAI indicator with both immigration and emigration, and for IDV (especially with immigration). The immigration percentage is again negatively correlated with LTO, yet stronger.

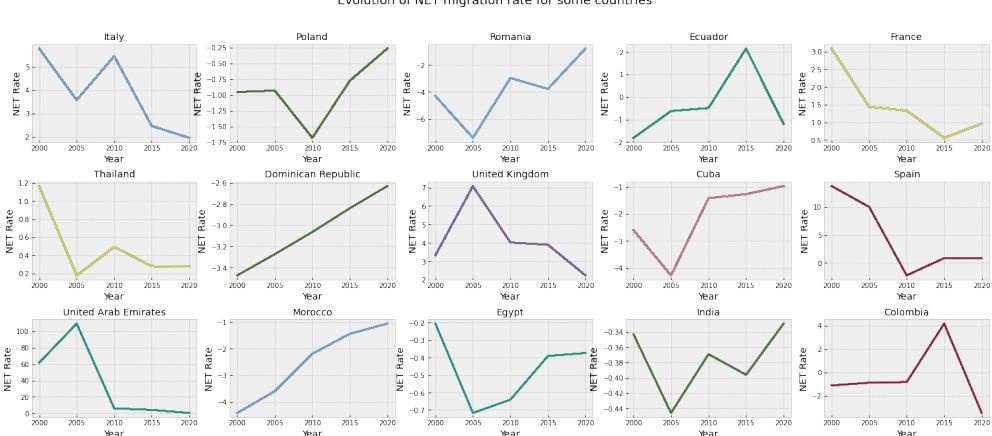

**Figure 9.** Evolution of five-year NET migration rate over time, for a sample of countries.

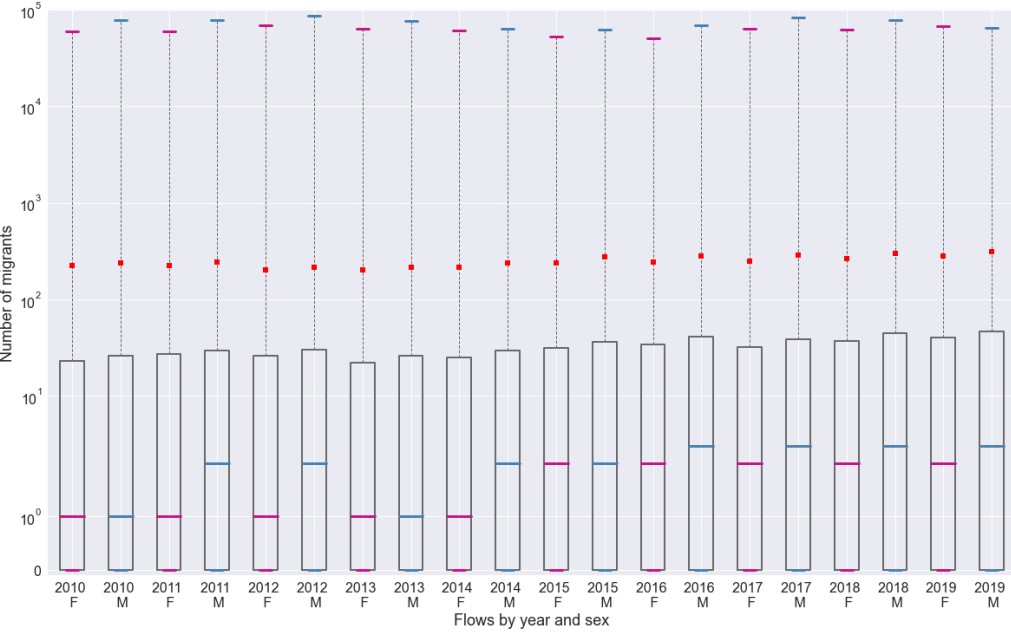

**Figure 10.** *Cont.*

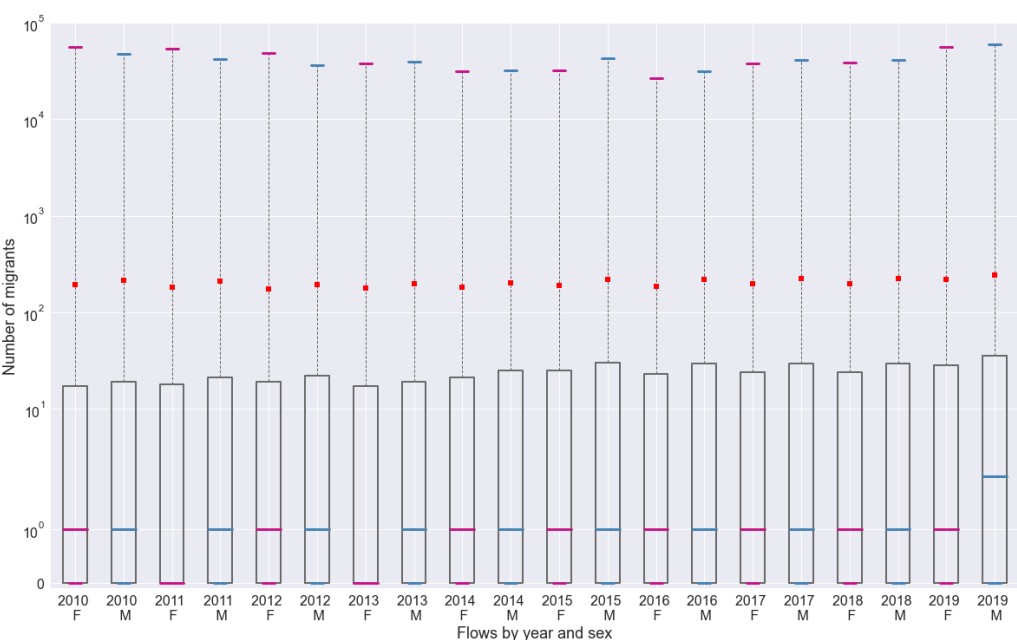

**Figure 10.** Distribution of migration flows by citizenship and by residence from EUROSTAT. Male migration is always higher than female migration for each annual measurement, while the average pattern over time is a slight increase in migration.

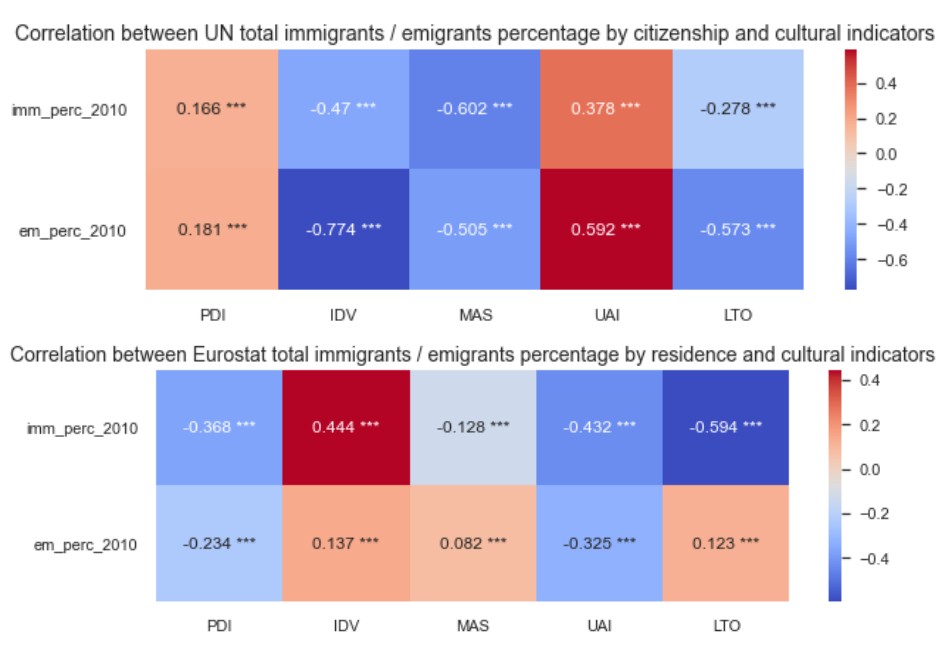

**Figure 11.** Correlation between cultural indicators and total immigrants/emigrants percentage from UN by citizenship and from EUROSTAT by residence. Absolute numbers of total migrants have been divided by the annual total population of the country, obtaining the annual percentage of total immigrants and emigrants. Positive values are presented in red while negative values in blue. Cultural indicators refer to 2008, while migration data is related to 2010. Asterisks indicate the significance levels of the correlation (***: *p*-value < 0.001).

The correlation between NET migration rate and GDP of a country shown in Figure 12 confirms the existing relation, well documented in literature, between these two variables. Correlation is always positive, meaning that countries with high GDP face a higher NET immigration and so confirming that high per capita income are conducive to migration [57]. Specifically, human migration is influenced by GDP values up to more than 10 years back.

**Figure 12.** Correlation matrix between annual GDP per capita and NET migration rate.

The heatmaps in Figure 13 illustrate the trends in linear correlations over years between EUROSTAT migration flows and migration stocks. Although the existing correlation between stocks at a given time $t$ and flows relative to previous years is self-evident (as those same flows will be included in the total counting of stocks), it is interesting to note that quite strong positive correlations also propagate forward in time: this could suggest that the higher the stock count at a given time $t$, the more migration flows will be shared by the pair of countries.

Finally, Figure 14 reveals existing correlations between Facebook SCI and the migration phenomenon measured through EUROSTAT and UN flows (by citizenship). Specifically, we have preliminary evidence of consistent positive values of Kendall's tau-b and Spearman rank-order correlations between SCI and international migration flows.

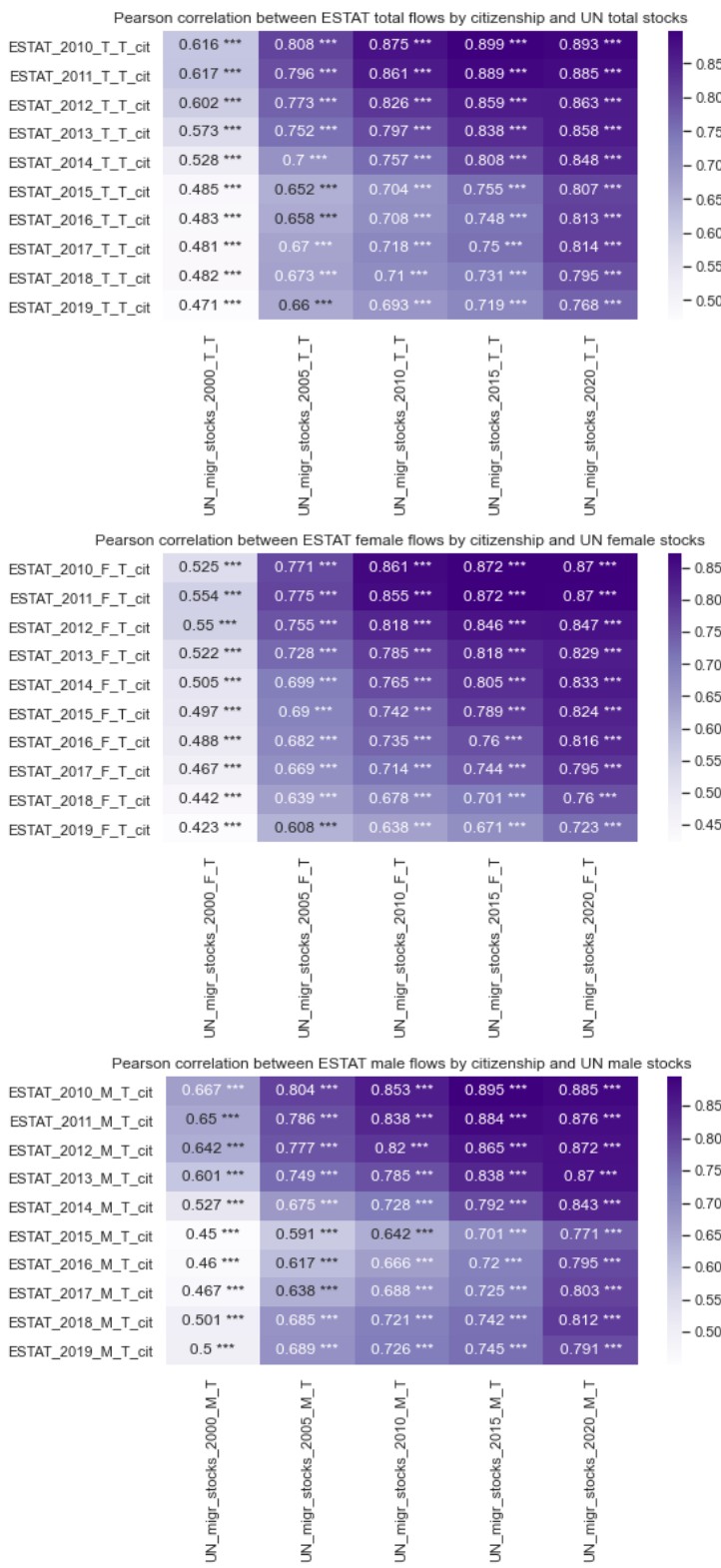

**Figure 13.** Correlation between migration stocks and ESTAT migration flows by citizenship, divided by sex.

| kendall | | |
|---|---|---|
| | sci_2020 | sci_2021 |
| ESTAT_2000_T_T_cit | 0.334 *** | 0.323 *** |
| ESTAT_2001_T_T_cit | 0.325 *** | 0.312 *** |
| ESTAT_2002_T_T_cit | 0.319 *** | 0.305 *** |
| ESTAT_2003_T_T_cit | 0.335 *** | 0.322 *** |
| ESTAT_2004_T_T_cit | 0.335 *** | 0.323 *** |
| ESTAT_2005_T_T_cit | 0.352 *** | 0.341 *** |
| ESTAT_2006_T_T_cit | 0.343 *** | 0.33 *** |
| ESTAT_2007_T_T_cit | 0.35 *** | 0.335 *** |
| ESTAT_2008_T_T_cit | 0.405 *** | 0.394 *** |
| ESTAT_2009_T_T_cit | 0.397 *** | 0.385 *** |
| ESTAT_2010_T_T_cit | 0.396 *** | 0.385 *** |
| ESTAT_2011_T_T_cit | 0.399 *** | 0.389 *** |
| ESTAT_2012_T_T_cit | 0.403 *** | 0.393 *** |
| ESTAT_2013_T_T_cit | 0.395 *** | 0.385 *** |
| ESTAT_2014_T_T_cit | 0.402 *** | 0.391 *** |
| ESTAT_2015_T_T_cit | 0.406 *** | 0.396 *** |
| ESTAT_2016_T_T_cit | 0.402 *** | 0.393 *** |
| ESTAT_2017_T_T_cit | 0.405 *** | 0.394 *** |
| ESTAT_2018_T_T_cit | 0.399 *** | 0.388 *** |
| ESTAT_2019_T_T_cit | 0.401 *** | 0.392 *** |
| UN_2010_T_T_cit | 0.313 *** | 0.319 *** |
| UN_2011_T_T_cit | 0.357 *** | 0.357 *** |
| UN_2012_T_T_cit | 0.376 *** | 0.379 *** |
| UN_2013_T_T_cit | 0.22 *** | 0.225 *** |
| UN_2014_T_T_cit | 0.142 *** | 0.144 *** |
| UN_2015_T_T_cit | 0.319 *** | 0.32 *** |
| UN_2016_T_T_cit | 0.243 *** | 0.242 *** |
| UN_2017_T_T_cit | 0.308 *** | 0.308 *** |
| UN_2018_T_T_cit | 0.315 *** | 0.322 *** |
| UN_2019_T_T_cit | 0.327 *** | 0.332 *** |
| UN_2020_T_T_cit | 0.209 *** | 0.218 *** |

| spearman | | |
|---|---|---|
| | sci_2020 | sci_2021 |
| ESTAT_2000_T_T_cit | 0.45 *** | 0.433 *** |
| ESTAT_2001_T_T_cit | 0.433 *** | 0.416 *** |
| ESTAT_2002_T_T_cit | 0.431 *** | 0.411 *** |
| ESTAT_2003_T_T_cit | 0.454 *** | 0.435 *** |
| ESTAT_2004_T_T_cit | 0.455 *** | 0.438 *** |
| ESTAT_2005_T_T_cit | 0.48 *** | 0.463 *** |
| ESTAT_2006_T_T_cit | 0.467 *** | 0.447 *** |
| ESTAT_2007_T_T_cit | 0.472 *** | 0.45 *** |
| ESTAT_2008_T_T_cit | 0.53 *** | 0.515 *** |
| ESTAT_2009_T_T_cit | 0.521 *** | 0.505 *** |
| ESTAT_2010_T_T_cit | 0.522 *** | 0.507 *** |
| ESTAT_2011_T_T_cit | 0.524 *** | 0.51 *** |
| ESTAT_2012_T_T_cit | 0.527 *** | 0.514 *** |
| ESTAT_2013_T_T_cit | 0.518 *** | 0.504 *** |
| ESTAT_2014_T_T_cit | 0.528 *** | 0.513 *** |
| ESTAT_2015_T_T_cit | 0.533 *** | 0.519 *** |
| ESTAT_2016_T_T_cit | 0.528 *** | 0.516 *** |
| ESTAT_2017_T_T_cit | 0.533 *** | 0.518 *** |
| ESTAT_2018_T_T_cit | 0.527 *** | 0.513 *** |
| ESTAT_2019_T_T_cit | 0.53 *** | 0.517 *** |
| UN_2010_T_T_cit | 0.43 *** | 0.439 *** |
| UN_2011_T_T_cit | 0.482 *** | 0.483 *** |
| UN_2012_T_T_cit | 0.492 *** | 0.497 *** |
| UN_2013_T_T_cit | 0.309 *** | 0.317 *** |
| UN_2014_T_T_cit | 0.199 *** | 0.202 *** |
| UN_2015_T_T_cit | 0.427 *** | 0.43 *** |
| UN_2016_T_T_cit | 0.337 *** | 0.335 *** |
| UN_2017_T_T_cit | 0.407 *** | 0.408 *** |
| UN_2018_T_T_cit | 0.418 *** | 0.426 *** |
| UN_2019_T_T_cit | 0.438 *** | 0.444 *** |
| UN_2020_T_T_cit | 0.281 *** | 0.293 *** |

**Figure 14.** Kendall's tau-b and Spearman rank-order correlations between SCI and migration flows by citizenship for both UN and EUROSTAT sources.

## 6. Conclusions

This work introduced MIMI, a dataset containing a heterogeneous collection of indicators for migration research. These include traditional flow and stock indicators from international organisations, multidisciplinary cultural, demographic and economic indicators, and the Facebook Social Connectedness Index, an indicator available from Meta. The main objective of publishing this dataset is to help overcome well documented issues in integrating data from disparate sources, traditional or not, to facilitate application of these data for modelling purposes by a wider range of researchers. For this purpose, we put together over 800 indicators for different years, uniforming locations, time scales, sources,

and notation. We describe the variables included in the data in detail and show different statistics, for descriptive and validation purposes.

The data obtained can be easily employed in nowcasting and forecasting studies, building models that integrate different variables, which is our next step in future work. We also plan to study the benefits and challenges of using non traditional data sources such as the Facebook SCI for prediction. While our current efforts concentrate on Facebook SCI, other non-traditional data types exist and could be integrated with our dataset, to enlarge its scope. For instance Twitter indicators on sentiment [58] or integration [14] could be added, Facebook Advertising indicators [59], mobile phone data [13], satellite data [60], air traffic data [61], and others [39,62].

**Author Contributions:** Conceptualization, D.G., L.P. and A.S.; dataset identification, D.G., L.P. and A.S.; data curation—preprocessing and integration, D.G.; validation, D.G., L.P. and A.S.; writing— original draft preparation, D.G.; writing—review and editing, D.G., L.P. and A.S.; supervision, A.S. All authors have read and agreed to the published version of the manuscript.

**Funding:** This work is supported by the European Union—Horizon 2020 Program through the project "HumMingBird—Enhanced migration measures from a multidimensional perspective", Grant Agreement n.870661, and under the scheme "INFRAIA-01-2018-2019—Integrating Activities for Advanced Communities", Grant Agreement n.871042, "SoBigData++: European Integrated Infrastructure for Social Mining and Big Data Analytics".

**Institutional Review Board Statement:** Not applicable.

**Informed Consent Statement:** Not applicable.

**Data Availability Statement:** The MIMI dataset v2.0 presented in this study is openly available on Zenodo at the following link: https://doi.org/10.5281/zenodo.6493325 (accessed on 21 August 2023).

**Acknowledgments:** We thank Jisu Kim, Stefano Maria Iacus and Spyridon Spyratos for useful discussions on dataset selection.

**Conflicts of Interest:** The authors declare no conflict of interest.

## Abbreviations

The following abbreviations are used in this manuscript:

GDP   Gross Domestic Product
PPP   Purchasing Power Parity
SCI   Social Connectedness Index
UN    United Nations

## Appendix A

We include below a detailed definition of all variables contained in our dataset and their sources.

- **Index** (*variable 1* in Table 2):
  The index consists in **uniquely identified pairs of countries**, built as follows: `ISO2 code of origin country—ISO2 code of destination country` (e.g., AL-FI index indicates records related to migration from Albania to Finland). The dataset contains also pairs having the same country codes for origin and destination, in order to include data on "returners" using migration flows by nationality or country of birth (e.g., BE-BE record will contain annual migration flows that represent people that were born or have citizenship in Belgium which moved their residence to Belgium in the reference year).
- **Facebook data** (*variable 2* in Table 2):
  This indicator uses anonymised insights of active Facebook users and their friendship networks to measure the intensity of connectedness between locations [16]. In this way, the resulting formulation in Equation (A1) is a measure of the social connectedness

between the two locations *i* and *j*, that is representative of the relative probability that two individuals across the two locations are friends with each other on Facebook.

$$SocialConnectednessIndex_{i,j} = \frac{FB\_Connections_{i,j}}{FB\_Users_i * FB\_Users_j} \quad \text{(A1)}$$

Specifically, in this work the concept of "locations" coincides with NUTS0 areas since our dataset only focuses on country-to-country bilateral migration. Nevertheless, SCI is also provided with respect to narrower geographical granularities, (e.g., NUTS2, NUTS3): we do not exclude future works focused on the study of migration trends at a smaller resolution (country-to-county, or county-to-county).

The SCI has a symmetric structure by definition of the concept of "friendship" and has been re-scaled to have a maximum value of 1,000,000,000 and a minimum value of 1. In our dataset, the minimum possible value was originally 0 (indicating pairs of countries for which the index was not available), subsequently replaced with an arbitrarily small value (chosen as half of the minimum available, i.e., 19) in order to fix problems when computing Pearson correlation of the logarithmic SCI.

- **Geographical variables** (*variables 3–15* in Table 2):

  - *variables 5, 6, 7, 8* are **ISO-3166 standards nomenclatures for country identification**, retrieved from PyCountry Python module [13] and ISAN (International Standard Audiovisual Number) [63].
  - *variables 9, 10* identify **continents** as follows: Africa (AF); Antarctica (AQ); Asia (AS); Europe (EU); America, North (AN); Oceania (OC); America, South (SA).
  - *variable 3* consists of the pair `code of origin continent`—`code of destination continent`. Its functionality can be fully appreciated in chord diagrams (examples in Section 5).
  - *variables 11, 12, 13* **indicate the position** of the centroids of both origin and destination countries in a classic geographic coordinate system. They are gathered and integrated from Google DSPL [64] and from the `latlng()` method of the CountryInfo Python library [14], and then merged together in a tuple (*variable 13*) built as a specific GeoPandas data structure called "geometry array" [15].
  - *variable 4* is the **measure of distance** between origin and destination, computed starting from the tuple in *variable 13* of both countries and using the geodesic formulation [16] [65] provided by the GeoPy Python library [17].
  - *variable 14* consists in the list of countries that **share a border** with the given country. The utility of this variable is to find out if the two countries of origin and destination share a border, using a straightforward function to check if a country name (*variable 6*) is contained into the list of neighbors of the other, and vice versa. An additional binary variable (e.g., "neighbors", having value `True` or `False`) could be derived from this method. Countries having empty list are islands. The corresponding sources for this variable are the following: GitHub repository in [66], `borders()` method of CountryInfo Python module [18] and Wikipedia [67].
  - *variable 15* is the **measure of the area** extension of the country in squared kilometers. It is gathered from The World Bank [68] and integrated with `area()` method of CountryInfo Python module [19].

- **Interdisciplinary indicators** (*variables 16–25* in Table 2):

  - *variable 17* is an indicator that provides per capita [20] annual values for **gross domestic product (GDP)** of a country, expressed in current international dollars and converted by purchasing power parity (PPP) [21] conversion factor. Data is retrieved from The World Bank [69]. The GDP is one of the "Development Indicators", already widely used in literature in combination with global migration [57].
  - *variables 16, 18* contain, respectively, the most practiced **religion**, and the list of the most spoken **languages** in the country (including both official and minority

languages). The benefit of including these columns would be to discover if the two countries of origin and destination share some languages or religions (or both), since this could favor a migratory exchange between the two. Rare languages and religions used only in one country and not shared with any other have been removed.Languages have been gathered from Wikipedia [70] while religions comes from DataHub [71] and have been integrated with Wikipedia data [72].

- *variables 19, 20* indicate the quantity (as absolute number and as percentage of the total population) of **Facebook users** that a given country has. The source is World Population review [73], which refers to the latest available measure for each country (oldest date back to December 2020).

- *variables 21–25* represent **Cultural Indicators** of a location, intended as dimensions along which cultural values of that location can be analysed [74]. Their origin dates back to the work of [75] although, over the decades, independent research branches led to the creation and addition of new ones [76]. Our work includes five of these indicators, of which we provide a brief individual description. Their applications in literature have been several (e.g., cross-cultural studies using Twitter data [77]), but the purpose of their inclusion in the MIMI dataset is to use them in an original way: our intention is to explore and understand their possible relation with international migration trends. Data about cultural indicators are available at different NUTS levels but in our work they only appear related to NUT0 (country) level since it is the only one that fits our geographic viewpoint. The values reported in the MIMI dataset are the result of the integration of two different datasets [78,79]. Unfortunately, they are provided only for 66 countries but, despite this, most of them have already shown to be strongly involved in migration trends (see also their correlation with the absolute number of migrants of a country, in Section 5.1). Starting from cultural dimensions of both countries of origin destination, a new variable about cultural distance could be obtained: datasets with this configuration already exist [74,80] although, at the moment, data is available only for a third of the countries (22 in total).

  * *variable 21* is the **Power distance indicator (PDI)** which is defined as "the extent to which the less powerful members of organisations and institutions (like the family) accept and expect that power is distributed unequally" [76]. This indicator describes the extent to which hierarchical relations and unequal distribution of power in organisations and societal institutions are accepted in a culture.

  * *variable 22*: the **Individualism indicator (IDV)** [22] (as opposed to collectivism) explores the "degree to which people in a society are integrated into groups" [76]: it reflects the extent to which people prefer to act as individuals rather than as members of a community.

  * *variable 23* is the **Masculinity indicator (MAS)**, defined as "a preference in society for achievement, heroism, assertiveness and material rewards for success" [76]. As opposed to femininity, this dimension reveals to what degree traditionally masculine societal values, such as orientation towards accomplishment, prevail over values such as modesty, solidarity or tolerance.

  * *variable 24* is the **Uncertainty avoidance indicator (UAI)** defined as "a society's tolerance for ambiguity", in which people embrace or avert an event of something unexpected, unknown, or away from the status quo [76].

  * *variable 25*: the **Long-term orientation indicator (LTO)** associates the connection of the past with the current and future actions/challenges. A lower degree of this index (short-term orientation) indicates that traditions are honoured and kept [76].

- **Demographic variables** (*variables 26–33* in Table 2):

- *variable 26*: annual **population stocks**, defined as the number of persons having their usual residence in a country in a given year, are gathered both from UN Population Division [81] (from which only records with "Zero migration" variant were selected) and EUROSTAT [82]: these two sources often refer to different groups of countries so their mutual integration allowed to cover most of the countries of the dataset. Where both measurements were available for the same country, both were reported. The two sources refer to different methodologies, since the annual total population measurement is performed on 1 July by UN, while on 1 January by EUROSTAT. However, their correlation value of ~1 proves that the two measures, related to the same year, are well compatible and almost interchangeable: indeed missing values related to the former have been replaced with the latter, and vice versa.

- *variable 27* represents annual **population density**, defined as the ratio between the annual average population and the land area. Therefore, its unit of measure corresponds to "persons per square kilometre". Data has been retrieved from ESTAT [83].

- *variables 28, 29*: **absolute number of migrants** (respectively, immigrants and emigrants) per country. Data was taken from ESTAT [84,85] and from UN datasets on flows (see below *variable 32*) selecting, from the latter, records having "Total" as country (respectively, origin and destination country). Regarding availability of variables for these two different sources, ESTAT total immigration and emigration are available only by residence, while UN total immigration is available both by citizenship and by residence, and UN total emigration is available only by citizenship. When records by citizenship were not available for some couples of countries they have been replaced with flows by county of birth. An additional indicator "total migration volume" [23], already existing and widely used in migrations studies, could be derived simply by summing *variables 28, 29*.

- *variables 30, 31* indicate quinquennial **NET migration** and **NET migration rate** of each country. The former is the difference between the number of immigrants and the number of emigrants in a given area during the reference year, while the latter is defined as the NET migration per 1000 persons and so it indicates the contribution of migration to the overall level of population change. A positive value for them indicates that there are more migrants entering than leaving a country (NET immigration), while a negative one means that emigrants are more than immigrants (NET emigration). Values have been taken from UN Population Division [86,87]: note that they apply also for EUROSTAT countries, and they have been widely used in literature in combination with them, even if NET migration rate calculation is based on midyear population (as required by the standard UN methodology).

- *variable 32*: yearly **migration flows** for each pair of countries are defined as the number of people that have moved country (i.e., that changed residence). Unlike a static stock measure, flow data are dynamic, summarising movements over defined periods and consequently allow for a better understanding of past patterns and the prediction of future trends [88]. Both EUROSTAT and UN divide migration flows into three categories: by residence [89–92], by citizenship [93,94] and by country of birth [95,96]. This is true in EUROSTAT for both inflows and outflows, while in UN only for inflows, as UN outflows exist only by residence. For our purposes, however, we selected EUROSTAT outflows only by residence, since the ones by citizenship and by country of birth cannot properly be defined as "flows", having a missing destination country. Additionally, this variable comprises, for both sources, migration flows by sex and by age and also by residence and by citizenship (for a total of 479 columns for EUROSTAT and 65 columns for UN). Similar to total migration (*variables 28, 29*), when records

by citizenship were not available for some couples of countries they have been replaced with flows by county of birth.

– *variable 33*: quinquennial **migration stocks** for each pair of countries consist in the absolute number of migrants residing in the destination country at a given time. Data is obtained from UN [97] and includes stocks by sex and age.

### Appendix B

In this appendix we include additional plots that can give further insight into the structure of the MIMI dataset.

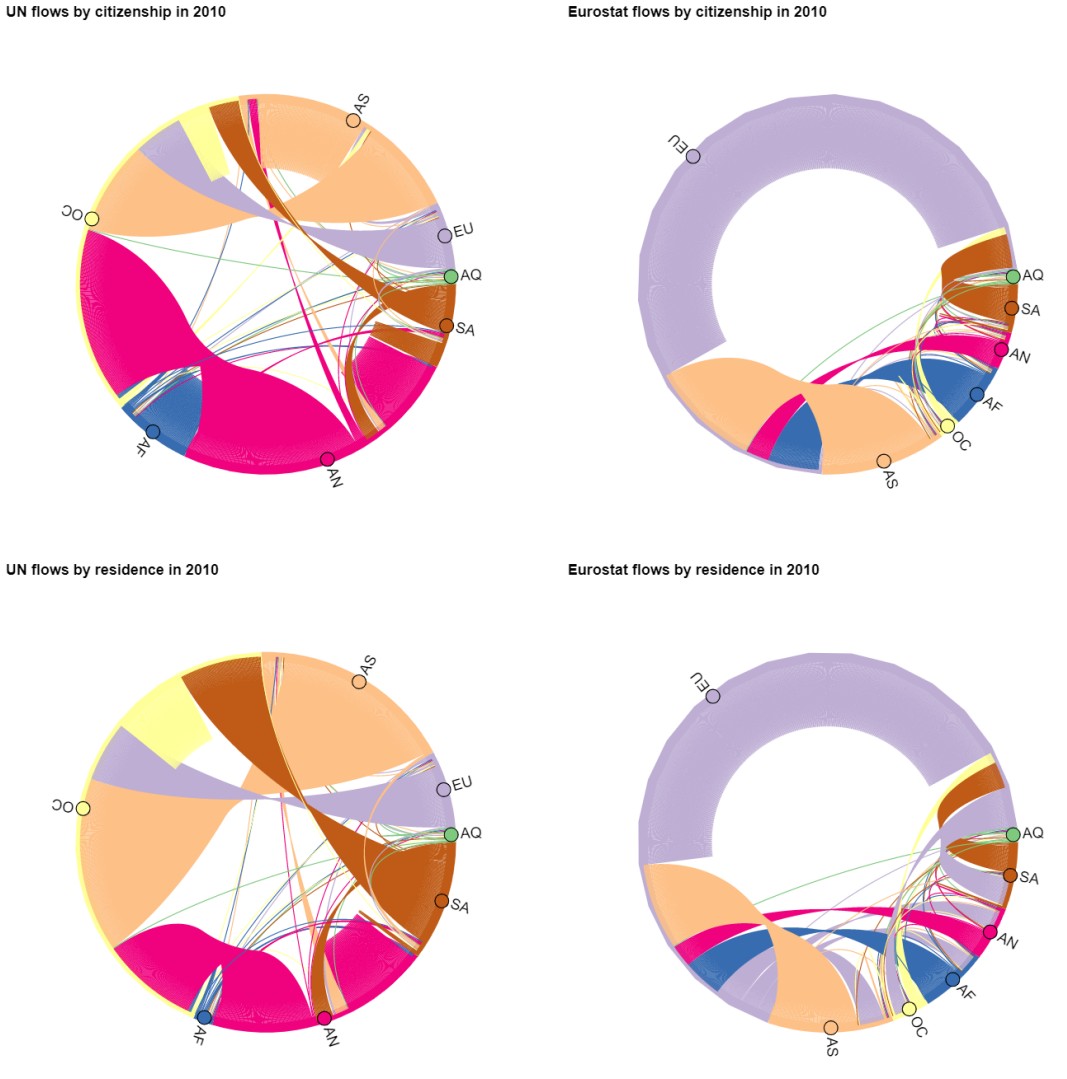

**Figure A1.** Inter-continental migration flows by citizenship and by residence from UN (**left**) and EUROSTAT (**right**) in 2010. Substantial immigration flows can be noticed for Oceania, having mainly America, Europe and Asia as origin continents. For EUROSTAT source, in addition to the main presence of internal flows, immigration from Asia is relevant.

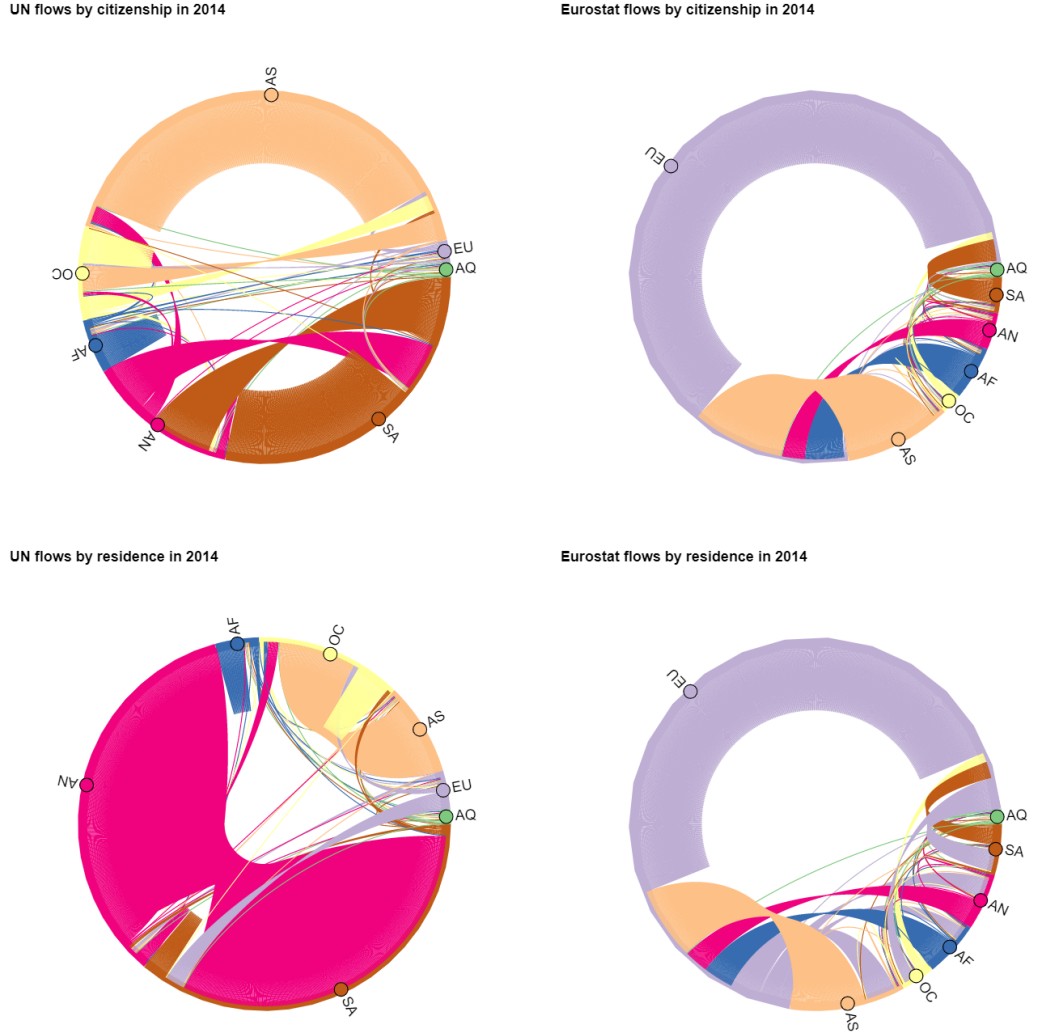

**Figure A2.** Inter-continental migration flows by citizenship and by residence from UN (**left**) and EUROSTAT (**right**) in 2014 . Compared to 2010 (figure above), immigration in Oceania has dropped, while bidirectional flows between North and South America have increased. EUROSTAT main trends remained unchanged.

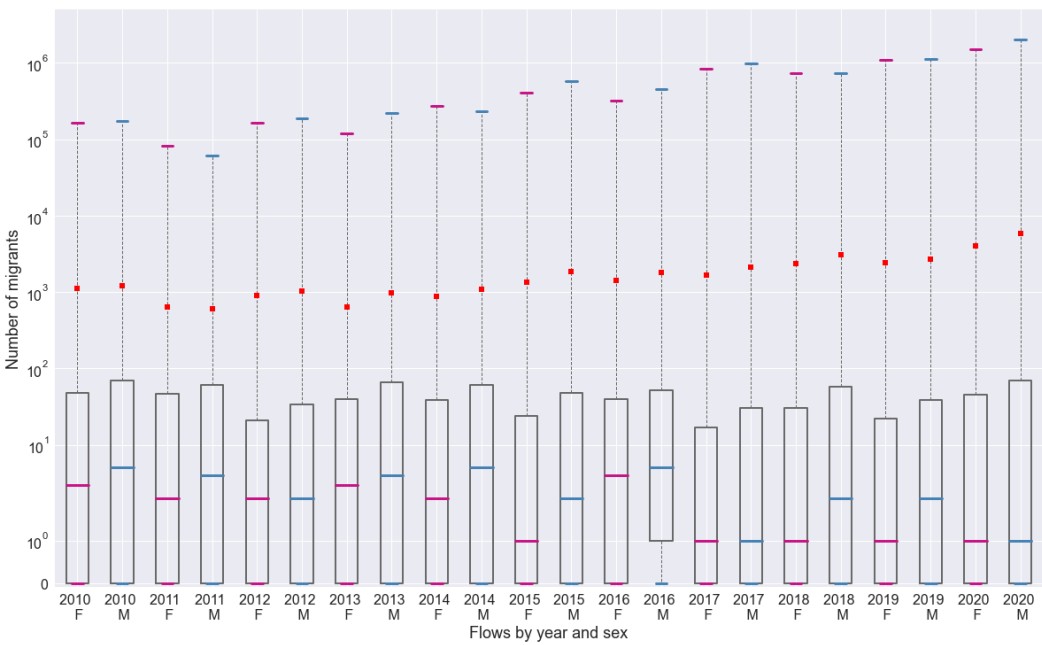

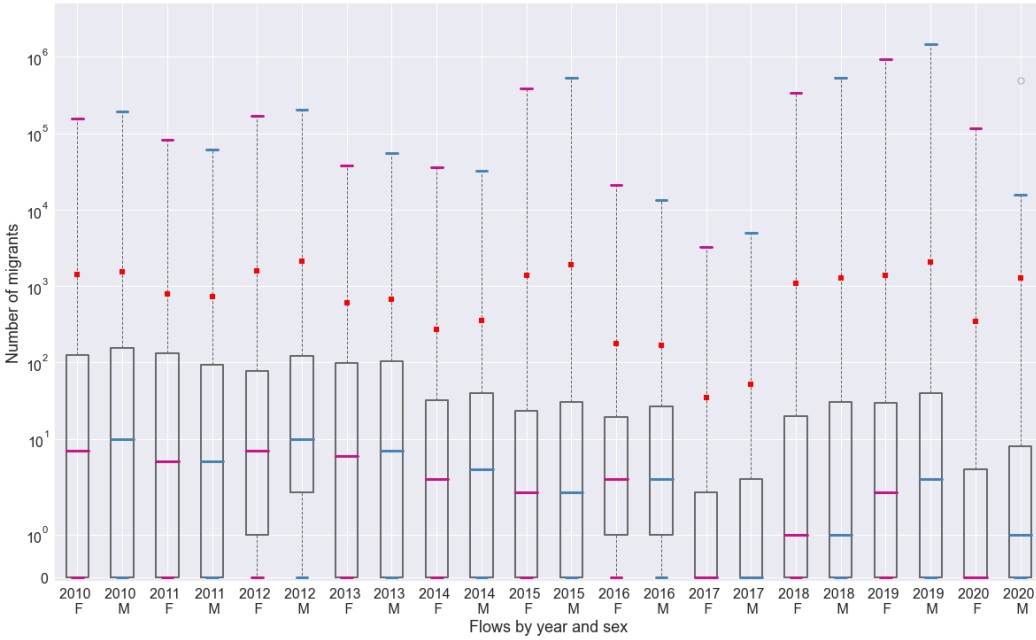

**Figure A3.** Distribution of migration flows by citizenship and by residence from UN. The increasing trend encountered in the previous figure is not present for these distributions, while the discrepancy between male and female migration still remains.

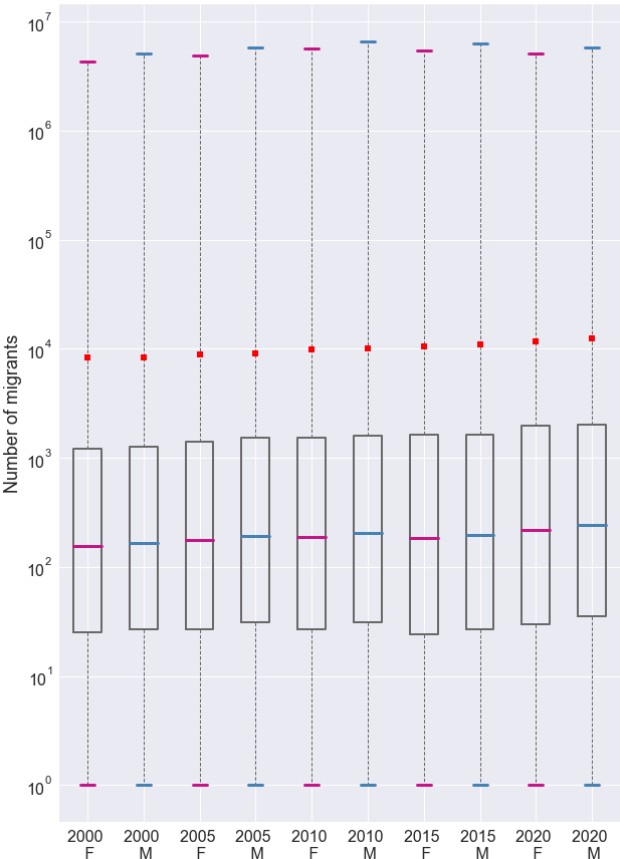

**Figure A4.** Distribution of migration stocks. The five year measurement prevents from having a more detailed look similar to the flows. Nevertheless, an increase in the general trend over years is quite evident.

## Notes

1    https://www.migrationdataportal.org/ (accessed on 21 August 2023)

2    i.e., developed regions, less developed regions, and the least developed countries.

3    Statista, a large online portal of world statistics, https://www.statista.com/ (accessed on 22 August 2023)

4    https://publications.iom.int/books/global-migration-indicators-2021 (accessed on 22 August 2023)

5    Dataset Providers includes initiatives that aim to "[..] create more data so that social impact organizations can create more data solutions so that social benefit increases" [45].

6    previously "Facebook Data for Good", https://dataforgood.facebook.com/ (accessed on 22 August 2023)

7    https://dataforgood.facebook.com/dfg/tools/social-connectedness-index (accessed on 22 August 2023)

8    https://creativecommons.org/licenses/by-nc/4.0/ (accessed on 21st August 2023)

9    https://bit.ly/Facebook_SCI (accessed on 21 August 2023)

10   https://bit.ly/DataForGoodAtMeta (accessed on 21 August 2023)

11   https://data.humdata.org/ (accessed on 21 August 2023)

12   https://jupyter.org/ (accessed on 21 August 2023)

13   https://pypi.org/project/pycountry/ (accessed on 22 August 2023)

14   https://pypi.org/project/countryinfo/#latlng (accessed on 22 August 2023)

15   https://geopandas.org/en/stable/docs/reference/api/geopandas.points_from_xy.html (accessed on 22 August 2023)

16   https://geopy.readthedocs.io/en/stable/#module-geopy.distance (accessed on 22 August 2023)

17   https://pypi.org/project/geopy/ (accessed on 22 August 2023)

18  https://pypi.org/project/countryinfo/#borders (accessed on 22 August 2023)
19  https://pypi.org/project/countryinfo/#area (accessed on 22 August 2023)
20  Calculated as the aggregate of production (GDP) divided by the population size.
21  a detailed definition of PPP provided by System of National Accounts 1993 Glossary can be found here: https://unstats.un.org/unsd/nationalaccount/glossresults.asp?gID=438 (accessed on 22 August 2023)
22  The same indicator can be found in other sources and contexts with the "IND" acronym which, however, can be confused with the Indulgence cultural indicator (IND).
23  https://ec.europa.eu/home-affairs/pages/glossary/total-migration_en (accessed on 22 August 2023)

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
