# Peer review of "Dataset of Multi-Aspect Integrated Migration Indicators"

_data, 2022_

Round 1

Reviewer 1 Report

The article „Dataset of Multi-aspect Integrated Migration Indicators“ is interesting and practically useful. The authors have tried to bring serious work to this paper. The method is adequately described. The results are clearly presented. The first section is a bit general and the reader cannot really see where the author(s) are and where they are going. If the authors rearrange and adopt a critical point of view when writing the theoretical framework, information will be then meaningful. The key to effective research is not just gathering a lot of information, but also evaluating the information and making sense of it. The author should work harder on the approach adopted, establish a clear theoretical background to contextualize the analysis, and narrow the scope of the analysis to specific aspects. The conclusions should be expanded. The purpose of the study and its theoretical and practical contributions should be mentioned. Suggestions for future work should be put forward in more detail.

Reviewer 2 Report

--Provide the nature of leading studies in order to distinguish between them and supporting studies,

--The research paradigm of the study which should provide the language of the study is not clearly articulate. It should be clarified in terms of its ontology, epistemology, axiology...;

--Most parts of the study are good.

Reviewer 3 Report

One novelty of this paper is that this dataset combines traditional and novel variables. However, the only "novel" variable is the Facebook data with only two snapshots in 2020 and 2021. This dataset contains migration information since 2010. Although the authors present correlation analysis at the end, I really doubt that how useful of the Facebook data as an indirect indicator of migration in earlier years. 

Additionally, the novelty of this dataset is the integration of traditional and novel variables. The authors should have more novel variables besides the Facebook data. 

Detailed comments:

1. I would suggest the authors change the organization of the paper. It seems like the whole paper is the description of the data set. The authors use 8 pages from page 2 to page 9 just describing all the variables from the dataset. This is a research paper, not a documentation of a dataset. The authors should explain why this dataset is important to the community and how useful it is. For example, the authors could use apply some migration models on this dataset to show that this is meaningful and helpful. 

2. The authors should add more related literature in the paper to discuss what are the existing datasets on migration indicators and how this is better than the existing work. 

3. In introduction, the authors write that "The new data have the advantage of timeliness and large geographical coverage, but also disadvantages in terms of selection bias and amount of resources required to process. Therefore, models extracted from these data need to be carefully validated, typically with traditional data sources." In this dataset, the authors use Facebook data as the new data. Does the Facebook data have the same problem? 

4. There are too many figures in the paper that are redundant and for some of them e.g. figure 6, 7, 8, 9, 10, 11 (third paragraph in 4.1) the authors never talk about the figure and discuss the takeaways from the figure. In the paper, figures should be used to help the authors tell a story. The authors need to discuss the takeaway from each figure and explain why this figure is important. 

5. Many figures are too small and hard to read e.g. figure 3, 4, 10. What is the y-axis scale in figure 13 and 14. Is it the number of people? The scale is too large from 0 to 10 million.

Round 2

Reviewer 3 Report

Thanks for the revision